# Tritium as hydrological tracer in Mediterranean precipitation events

Tobias R. Juhlke[1], Jürgen Sültenfuß[2], Katja Trachte[3], Frédéric Huneau[4,5], Emilie Garel[4,5], Sébastien Santoni[4,5], Johannes A. C. Barth[1], Robert van Geldern[1]

[1]GeoZentrum Nordbayern, Department Geographie und Geowissenschaften, Friedrich-Alexander-Universität Erlangen-Nürnberg, Schlossgarten 5, 91054 Erlangen, Germany
[2]Institut für Umweltphysik, Universität Bremen, Otto Hahn Allee 1, 28355 Bremen, Germany
[3]Institute for Environmental Sciences, Brandenburg University of Technology (BTU), Cottbus-Senftenberg, Germany
[4]Université de Corse Pascal Paoli, Faculté des Sciences et Techniques, Département d'Hydrogéologie, Campus Grimaldi, BP 52, F-20250 Corte, France
[5]CNRS, UMR 6134 SPE, F-20250 Corte, France

*Correspondence to*: Tobias R. Juhlke (tobias.juhlke@fau.de)

**Abstract.** Climate models are in need of improved constraints for water vapor transport in the atmosphere and tritium can serve as a powerful tracer in the hydrological cycle. Although general principles of tritium distribution and transfer processes within and between the various hydrological compartments are known, variation on short timescales and aspects of altitude dependence are still under debate. To address questions for tritium sources, sinks and transfer processes, sampling of individual precipitation events in Corte on the island of Corsica in the Mediterranean Sea was performed between April 2017 and April 2018. Tritium concentrations of 46 event samples were compared to their moisture origin and corresponding air mass history. Air mass back-trajectories were generated from the novel high-resolution ERA 5 data set of the ECMWF (European Centre for Medium-Range Weather Forecasts). Geographical source regions of similar tritium concentrations were predefined using generally known tritium distribution patterns, such as a 'continental effect', and from data records derived at long-term measurement stations of tritium in precipitation across the working area. Our model-derived source region tritium concentrations agreed well with annual mean station values. Moisture that originated from continental Europe and the Atlantic Ocean was most distinct regarding tritium concentrations with values up to 8.8 TU and near 0 TU, respectively. Seasonality of tritium values ranged from 1.6 TU in January to 10.1 TU in May and exhibited well-known elevated concentrations in spring and early summer due to increased stratosphere-troposphere exchange. However, this pattern was interrupted by extreme events. The average altitude of trajectories correlated with tritium concentrations in precipitation, especially in spring and early summer and if outlier values of extreme tritium concentrations were excluded. However, in combination with the trajectory information, these outlier values proved to be valuable for the understanding of tritium movement in the atmosphere. Our work shows how event-based tritium research can advance the understanding of its distribution in the atmosphere.

*Keywords*: hydrologic cycle, tritium, isotope hydrology, air mass trajectory, atmospheric moisture transport, precipitation events, HYSPLIT

# 1    Introduction

The hydrologic cycle is one of the key components in today's climate models that predict the evolution of climate parameters in the face of climate change. An improved constraint of model input parameters such as atmospheric moisture transport has gained increased scientific interest (Flato et al., 2013). For instance, cloud formation (Seinfeld et al., 2016) and planetary boundary layer (PBL) representation (Teixeira et al., 2008) are considered some of the main sources of error in model scenarios. In addition to computational advances, measurements of hydrochemical parameters can serve to better constrain

moisture transport patterns. Ideal parameters for this task should therefore directly trace the movement of the water molecule itself.

One widely used tracer for this is tritium ($^3$H or T) that enters the hydrological cycle mostly in the form of tritiated water (HTO). Tritium is a radiogenic isotope of hydrogen with a half-life of 12.32 (±0.02) years (Lucas and Unterweger, 2000), that has natural and anthropogenic sources to the environment.

Natural tritium is produced by cosmic radiation that interacts with nitrogen in the upper atmosphere (Craig and Lal, 1961). Since the natural production of tritium as well as its decay is constant over time, there is a constant global inventory of natural tritium. The average concentration without anthropogenic influence is about 5 TU (1 TU = $^3$H/$^1$H ratio of $10^{-18}$) in precipitation for Northern Europe (Roether, 1967). Into this natural background, fusion bomb tests, conducted until the early 1960s, added high concentrations of tritium to the environment. This resulted in peak concentrations of tritium in the hydrosphere in 1963

(GNIP station in Vienna; IAEA/WMO, 2019). This tritium bomb-peak was used in various studies as a method to trace and date groundwater (Von Buttlar and Wendt, 1958; Allison and Holmes, 1973), to quantify recharge (Vogel et al., 1974; Allison and Hughes, 1975) or to investigate ocean circulation (Jenkins and Rhines, 1980; Fine et al., 1981). The generally observed decline of tritium concentration in precipitation since peak times in the 1960s is the result of multiple processes. Due to radioactive decay elevated concentrations decrease over time. Additionally, intermediate storage of tritium in groundwater,

absorption by plants, and long-term storage in deep ocean water lowers the tritium amount in the more active part of the hydrologic cycle. Current anthropogenic release of tritium into the environment is a byproduct of nuclear facilities, such as nuclear power and reprocessing plants. Emissions from these sources are several magnitudes lower than historical bomb-derived tritium concentrations. In the past, natural variations of tritium in meteoric waters were masked by input of bomb-generated tritium (Palcsu et al., 2018). The latter has now decayed to concentrations near levels, which allows investigations

into the natural tritium cycle. Natural levels are expected to be reached during the next decade (Eastoe et al., 2012). Thus, a thorough assessment of these variations is necessary for closer understanding of tritium distribution in recent and future studies to map natural tritium distributions and delineate sources and sinks in the hydrologic cycle (Cauquoin et al., 2016; Cauquoin et al., 2015).

A conceptual model of tritium with exclusively natural origin would have the upper atmosphere as the only source of tritium.

Through rainout, tritium would reach the groundwater where it is stored and eventually discharged to streams and the ocean. In the ocean, any continental runoff or direct precipitation becomes diluted in the much larger volume of ocean water. The

larger timescales of ocean circulation prevent a buildup of the decaying tritium. Thus, tritium concentrations of ocean water and its evaporated vapor are expected to be near zero. During air mass transport, direct uptake of high-tritium vapor from the upper atmosphere and re-evaporation control the distribution of tritium in precipitation that is collected on land. Comparable

to the stable isotope ratios of oxygen and hydrogen in the water molecule, several 'effects' may affect the distribution of tritium in precipitation. The 'latitude effect' shows increasing tritium concentrations with increasing latitude (Schell et al., 1974), the 'seasonal effect' exhibits increasing tritium concentrations during spring and early summer (Libby, 1963), and the 'continental effect' describes rising tritium concentrations in precipitation with increasing distance from coasts (Schell et al., 1970). Additionally, stratospheric moisture contains tritium contents of one (Ehhalt, 1971) to several magnitudes (Ehhalt et al., 2002)

higher than moisture near ground level. Even small amounts of this moisture can substantially increase tritium concentrations of resulting precipitation (Aggarwal et al., 2016). Since the exchange between stratosphere and troposphere varies geographically and over time (Jordan et al., 2003), the dynamics of stratospheric tritium contribution remains a partly open question (Cauquoin et al., 2015).

All these effects have been observed in monthly integrated samples of precipitation (Mook et al., 2001) and predicted by

models (Cauquoin et al., 2015; Visser et al., 2018). However, they currently lack a thorough investigation on single precipitation event basis. To better understand short term tritium variations in precipitation, air mass back-trajectory modelling was applied here to identify sources of water vapor for individual precipitation events and in order to compare resulting tritium concentrations with their expected origin. In particular, we aimed to compare spatial tritium distribution patterns from the Global Network of Isotopes in Precipitation (GNIP; IAEA/WMO, 2019) with moisture source regions of discreetly sampled

events. This comparison also had the purpose to assess influences of sampling on different timescales. Additionally, we examined correlations between tritium concentrations and altitude history of air masses to outline contributions of upper atmosphere tritium to single precipitation events. The improved resolution of available meteorological grid datasets in the form of the new ERA 5 dataset (Copernicus Climate Change Service (C3S), 2017) from the European Centre for Medium-Range Weather Forecasts (ECMWF) was an important prerequisite for our high-resolution study. Another important aspect of this

study is the location of sample collection in Corsica in the Mediterranean. In the Mediterranean multiple origins of air masses are observed throughout the year (Thiébault and Moatti, 2016). Precipitation samples with varying moisture origin are crucial for an in-depth comparison of multiple source regions. Our results will enhance the understanding of natural tritium dynamics and moisture movement in the atmosphere.

## 2    Study site and methods

Precipitation samples were collected between April 2017 and April 2018 in the city of Corte on the Mediterranean island of Corsica, France (42.300570° N, 9.148592° E, 415 m a.s.l., Figure 1). It is located in central Corsica, on the eastern part of the island's main mountain ridge that reaches an elevation of up to 2706 m a.s.l. East of Corte a mountain range, named

Castagniccia, is situated with an elevation of up to 1767 m a.s.l. Corte lies on the west wind lee side of the main mountain ridge, and also because it is not directly situated at the sea, a more diverse moisture origin for precipitation events is expected.

The transport of water vapor in the western Mediterranean is mainly controlled by atmospheric weather systems, which cause a strong seasonality in its occurrence and origin. During the boreal winter season Corsica is generally affected by the large scale circulation patterns related to the Northern Atlantic Oscillation (NAO) as well as Arctic Oscillation (AO) and accompanied dominant westerly wind regimes (Dünkeloh and Jacobeit, 2003). Particularly the windward side of the island, i.e. western slopes experiences air masses originating over the Atlantic Ocean, while the eastern slopes are mostly in the

leeward side. Precipitation thus mainly occurs from September until April when extra-tropical cyclones passes the island. The atmospheric forcings change in the summer season to weak synoptic conditions. This enables the development of local breeze-systems in the diurnal cycle, e.g. sea-slope breezes, which induce modifications in the vapor transport along altitudinal gradients towards the inland of Corsica (Burlando et al., 2008). The result is the formation of rather local precipitation events in the highlands. At coastal areas the descending branch of the local circulation cell from the slope breezes tends to suppress

convective activities and induce dry conditions from mainly May until August.

For tritium analyses, 1 L samples of precipitation were collected for 42 events. This number is a subset of all precipitation events that is expected to be representative of the total rainfall during the study period. In order to achieve this sample volume also from small precipitation events, a canvas tarp was laid out for collection of rooftop water. Samples were transferred from the collector bucket to 1 L bottles directly after the rain event. The sample bottles were then shipped to the Department of

Oceanography of the Institute of Environmental Physics (IUP) at the University of Bremen for analyses. Tritium concentrations were analyzed using the $^3$He-ingrowth method (Clarke et al., 1976). Samples were divided into two aliquots of 500 mL, degassed and stored in dedicated helium-free glass bulbs for the accumulation of the tritium decay product $^3$He. After a period of two to three months, the $^3$He content was analyzed by noble gas spectrometry. Details on the instrument setup can be found in Sültenfuß et al. (2009). All laboratory results of tritium concentrations were corrected for radioactive decay back to the time

of the precipitation event. Concentration values are reported in tritium units (TU), where 1 TU equals a radioactivity concentration of 0.118 Bq L$^{-1}$. This analytical setup allows a precision of ±3 % and a detection limit of 0.02 TU.

In total, 46 samples were analyzed that represent 42 single rain events on different days. The discrepancy between sample and event count is due to a chronological division of three precipitation events into multiple sub-samples in order to identify possible moisture origin changes during longer rain events.

**2.1    HYSPLIT trajectory model**

The Hybrid Single-Particle Lagrangian Integrated Trajectory model (HYSPLIT) of the National Oceanic and Atmospheric Administration (NOAA) Air Resources Laboratory (Rolph et al., 2017; Stein et al., 2015) was used to identify the origin of air masses at the precipitation sampling site during sampled rain events. With the model's meteorological information it is possible to identify locations where air masses gained and lost moisture along their travel paths to the study site.

The HYSPLIT model requires input in the form of meteorological grid data. We used the newly available ERA5 grids (Copernicus Climate Change Service (C3S), 2017) from the European Centre for Medium-Range Weather Forecasts. This dataset originally has a horizontal resolution of 31 km on a global scale, a vertical resolution of 37 interpolated pressure levels and a temporal resolution of one hour. As input for this HYPLSIT analysis a data subset was extracted with the following specifications: The horizontal and temporal resolutions were left at 31 km and 1 hour. The spatial extent of our model area

was clipped to an area of about 40 ° latitudinal and longitudinal distance from our sampling location (0 °N to 80 °N, 31 °W to 49 °E). The upper 6 pressure levels were omitted, leaving 31 pressure levels from 1000 to 20 hPa. Additionally, the model run was cropped in HYSPLIT to an elevation limit of 10000 m a.s.l. HYSPLIT tracks air parcels from a given four-dimensional coordinate (start time, start location, and start altitude) backwards in time. The result is a trajectory of spatial points at hourly time intervals. Additional meteorological parameters, such as specific humidity (SH) are attached to each point. The run time

of the model determines the length of each trajectory.

The location that served as input corresponds to the coordinates of our sampling site (see above). To track the specific air mass that results in cloud formation at the sampling site, the model input for elevation should ideally correspond to the cloud altitude at the sampling site. The cloud altitude at the sampling site was not measured, and we therefore performed twelve model runs for elevations from 0 to 6000 m above ground level in steps 500 m each. To cover the time period of a rain event, we started

the model on each full hour during a rain event. The run time was set to 10 days backward from the time of the precipitation event. Trajectories that leave the spatial domain specified by the meteorological input data (see above) were truncated because accuracy of trajectories after long distances and especially on long time scales (> 10 days) were assumed to be too low to allow a satisfactory interpretation. An example rain event of four hours duration thus results in 48 trajectories (four start times with twelve start elevations) with 240 hourly spaced points each, when assuming they stay within the model boundaries. In order

to test for the influence of trajectory backward runtime on the comparison with tritium concentrations we repeated the moisture source calculation for shorter durations of 3, 5 and 7 days. For consistency, all figures display the maximum 10-day-trajectory results. Differences between different trajectory run times, if significant, are discussed in the text where appropriate.

## 2.2    Moisture source identification

In order to identify moisture sources for the sampled rain events, meteorological parameters from the calculated HYSPLIT

trajectories were applied. The basic idea is that changes in specific humidity, which is one of the output parameters of the HYSPLIT model run, show locations where the air parcel took up or gave away moisture from its surroundings. This general concept has already been widely used with some small modifications to the calculation procedure (Pfahl and Wernli, 2008, 2009; Sodemann et al., 2008).

In this study, we used the approach of Visser et al. (2018) with minor adjustments to locate source areas of moisture uptake

for the sampled rain events. For each point of all trajectories three numerical values (weights) are calculated, whereas the product of those three weights represents the relative moisture contribution of that time and point to the precipitation event moisture. The first weight $w_r$ is used to determine which hours of the precipitation event have more influence on the rain

sample composition. Therefore, all points of all trajectories that start at the same hour receive the same weight $w_r$. It is calculated as the proportion of the precipitation rate at the trajectory start hour to the total precipitation amount of the corresponding event. The second weight $w_z$ is used to disentangle the contribution of the different trajectory start altitudes to the event. All points of a single trajectory receive the same weight $w_z$. The assumption is that the difference in water content between the last two points of a trajectory leading up to its termination at the sampling site is representative of this trajectory altitude's relative moisture contribution. The first step to construct $w_z$ is calculating the difference in specific humidity (SH) between the last and second to last trajectory point. If this difference is positive (increase in humidity during the last step) it is set to zero. Then this decrease in moisture is divided by the sum of SH decrease during the last hour of all trajectories with the same start time, which equals $w_z$. The third weight $w_{incr}$ reflects the water uptake along a trajectory, resulting in different values for each trajectory point. The calculation procedure for this weight uses a stepwise summation of SH differences from oldest to newest point of a trajectory. So an increase of SH from one trajectory point to the next denotes moisture uptake and is assigned to this trajectory point. A decrease of SH between two points corresponds to a loss from the present total amount of moisture. Therefore, the weights of all previous contributing trajectory points (moisture sources) are proportionally reduced, because the lost moisture is assumed to represent the well-mixed pool of previous moisture sources. At the end of this stepwise calculation procedure all weights are divided by the SH value of the final trajectory point in order to give the proportional contribution. Thus, the sum of these final $w_{incr}$ values along one trajectory sum up to one. The combined weight $w_s$ after which moisture sources are evaluated is calculated by multiplication of the three previous weights $w_r$, $w_z$, and $w_{incr}$ for each trajectory point. All values for $w_s$ of a single precipitation event sum to one, and therefore each one denotes the fraction of moisture that originates from a single trajectory point. For a more detailed explanation of these calculation procedures, including figures and equations, we refer to the Methods section of Visser et al. (2018).

Two examples for trajectories calculated for a precipitation event are shown in Figure 1. Example (a) from mid-December 2017 shows an air mass origin from Northern Atlantic and Polar regions, with main moisture uptake above the northeastern Atlantic Ocean and the northeastern Mediterranean Sea. Example (b) from the beginning of November 2017 shows air masses that took a different route, via the Strait of Gibraltar, and other parts that arrive from the southern Mediterranean. Here the main moisture source is the southern to western Mediterranean Sea.

[Figure 1 near here]

## 2.3    Tritium in moisture source regions

In order to investigate the tritium content of air masses of different origin, trajectory points were assigned to one of five different geographical regions (Figure 2). These five regions were delineated based on expected similarity in tritium values from the currently known tritium formation and distribution processes. This region classification was verified by the annual tritium average of GNIP stations (Figure 2). Western Europe shows lower values than the more eastern, continental parts. Further, the assumption of low values in marine moisture makes it plausible to delineate the Atlantic Ocean and the Mediterranean Sea from land masses. Africa was singled out as a rare, continental source area of moisture for the trajectories

investigated. This is plausible because predominant wind directions in the investigation are expected from the west and north (Thiébault and Moatti, 2016).

[Figure 2 near here]

In this simplified model of distinct tritium source areas, the final tritium value of a precipitation event was expected to depend on the amount of moisture that originates from a certain moisture region. This leads to the equation of a linear mixing model

$$TU_{\text{event}} = \sum_{n=1}^{5} (TU_n \cdot w_{s\,n})$$

(1)

where $TU_{\text{event}}$ is the tritium concentration of the rain event, $TU_n$ is the tritium concentration of one of the five source regions, and $w_{s\,n}$ is the relative moisture contribution of the source regions to the precipitation event. The tritium concentration of 46 precipitation events was measured and the relative moisture contribution of the five source areas was calculated from the trajectory analyses. For further calculations, it was assumed that $TU_n$ values of source areas were constant over time and greater than zero. Potential seasonal trends in tritium values will be discussed later. This leads to 46 equations with five unknowns each. The solution of this overdetermined system of equations can be approximated by a least squares approach, in which the sum of all squared deviations of predicted to measured values is minimized.

## 3    Results

Results of laboratory analyses of event water samples can be found in Table 1 and are also archived in the World Data Center PANGAEA[1]. Tritium values range between 1.62 and 10.07 TU with a mean value of 4.52 TU. A seasonal distribution with a spring to early summer maximum can be observed.

---

[1] www.pangaea.de; DOI: 10.1594/PANGAEA.911474

**Table 1.** Values for precipitation amount and tritium concentration of all measured precipitation event samples. Letters after tritium concentration refer to the discussion of outlier values in Sect. 4.3.

| Date | precipitation amount /mm | Tritium concentration /TU |
|---|---|---|
| 2017-04-18 | 0.2 | 7.46 |
| 2017-04-27 | 0.8 | 3.26 |
| 2017-04-27 | 1.0 | 3.64 |
| 2017-04-27 | 4.0 | 4.48 |
| 2017-05-03 | 0.4 | 4.00 |
| 2017-05-04 | 1.4 | 4.71 |
| 2017-05-04 | 0.4 | 5.69 |
| 2017-05-06 | 0.9 | 6.18 |
| 2017-05-06 | 3.0 | 3.71 |
| 2017-05-08 | 2.6 | 8.55 (A) |
| 2017-05-19 | 1.0 | 10.07 (B) |
| 2017-06-01 | 0.8 | 4.93 |
| 2017-06-02 | 0.2 | 8.10 (C) |
| 2017-06-03 | 0.8 | 3.93 |
| 2017-06-05 | 3.0 | 5.50 |
| 2017-06-28 | 2.2 | 5.18 |
| 2017-06-30 | 2.6 | 4.96 |
| 2017-07-15 | 7.8 | 5.62 |
| 2017-07-26 | 10.2 | 5.61 |
| 2017-09-08 | 1.4 | 2.82 |
| 2017-09-10 | 4.0 | 3.44 |
| 2017-09-15 | 0.2 | 2.24 |
| 2017-09-18 | 10.1 | 2.88 |
| 2017-11-05 | 6.7 | 4.38 |
| 2017-11-06 | 1.2 | 4.14 |
| 2017-11-09 | 0.2 | 3.29 |
| 2017-11-10 | 3.2 | 3.22 |
| 2017-11-13 | 0.8 | 8.85 (D) |
| 2017-11-29 | 0.8 | 3.17 |
| 2017-12-02 | 15.2 | 2.58 |
| 2017-12-08 | 0.2 | 2.83 |
| 2017-12-10 | 2.0 | 2.06 |
| 2017-12-14 | 0.1 | 1.80 (E) |
| 2017-12-16 | 2.8 | 2.79 |
| 2017-12-17 | 2.7 | 4.76 |
| 2017-12-18 | 1.4 | 4.11 |
| 2018-01-08 | 0.1 | 1.62 (F) |
| 2018-02-03 | 18.0 | 3.90 |
| 2018-02-06 | 13.2 | 3.79 |
| 2018-02-21 | 4.4 | 5.28 |
| 2018-02-22 | 2.2 | 5.23 |
| 2018-02-24 | 27.2 | 3.99 |
| 2018-03-02 | 12.6 | 4.26 |
| 2018-03-21 | 8.0 | 4.56 |
| 2018-03-31 | 6.6 | 3.82 |
| 2018-04-04 | 6.0 | 6.36 |

The relative contribution of moisture to the 46 precipitation samples that originated from the five tritium source regions was calculated after Eq. (1) and displayed in Figure 3. The average moisture contribution of each region is shown in Table 2. Most of the moisture at the sampling site, around 60 %, has its origin in the Mediterranean Sea. Western Europe and the Atlantic Ocean are the other two major contributors with around 10 to 15 % moisture contribution. Continental Europe and Africa show only minor contribution with values from 10 % (mean) to 0 % (median). Relative contributions for some of these regions are

not normally distributed, which results in the deviation of mean from median values. This is especially the case for the regions Africa and Continental Europe, where the dataset is skewed because they do not contribute any moisture for most events. The mean and median moisture contribution of the constructed tritium source regions does not vary considerable with the chosen trajectory backward run time (Table 2) of 3, 5, 7, and 10 days.

[Figure 3 near here]

The results of the tritium model in source regions (Table 2) after Eq. (1) show the highest tritium concentrations in Continental Europe with around 8.8 TU, followed by Western Europe with 7.3 TU. The negative value for the Atlantic Ocean is a result of the computation algorithm and indicate that a general trend towards minor tritium concentrations is the best fit to measurements. Dismissing the negative values of modelled tritium concentration for the Atlantic Ocean, the range of tritium values introduced by different trajectory run times is mostly less than 1 TU. For Western Europe the 10-day-trajectories show

a notably higher tritium concentration of 7.3 TU.

**Table 2. Results for the relative moisture contribution and the estimation of tritium in source regions (as defined in Figure 2) as calculated by Eq. 1 for different trajectory backward run times.**

| Tritium source region | Mean moisture contribution to precipitation events % | | | | Median moisture contribution to precipitation events % | | | | Estimate of tritium concentration TU | | | | Standard error of estimate of tritium concentration TU | | | |
|---|---|---|---|---|---|---|---|---|---|---|---|---|---|---|---|---|
| For days of trajectory backward run time | 3 | 5 | 7 | 10 | 3 | 5 | 7 | 10 | 3 | 5 | 7 | 10 | 3 | 5 | 7 | 10 |
| Africa | 6 | 6 | 6 | 5 | 0 | 0 | 0 | 0 | 5.50 | 4.56 | 4.80 | 5.52 | 1.85 | 1.77 | 1.81 | 1.97 |
| Atlantic Ocean | 8 | 11 | 11 | 12 | 4 | 9 | 10 | 9 | -0.14* | 1.39 | 0.12 | -1.21* | 2.58 | 2.42 | 2.79 | 2.74 |
| Continental Europe | 10 | 10 | 11 | 10 | 0 | 0 | 0 | 1 | 7.84 | 8.15 | 8.72 | 8.81 | 1.37 | 1.36 | 1.32 | 1.56 |
| Mediterranean Sea | 61 | 58 | 57 | 57 | 64 | 60 | 59 | 60 | 4.34 | 4.54 | 4.23 | 4.14 | 0.62 | 0.76 | 0.74 | 0.73 |
| Western Europe | 16 | 15 | 14 | 15 | 13 | 14 | 13 | 14 | 5.10 | 4.20 | 5.82 | 7.34 | 1.81 | 2.53 | 2.74 | 2.50 |

* For negative values see results and discussion


## 4 Discussion

### 4.1 Basic trends in tritium concentrations

The measured tritium values in precipitation at Corte are well within the range of observed tritium variances measured by European GNIP stations. A Shapiro-Wilk test reveals that the tritium concentrations of precipitation events are normally distributed ($p = 0.0053$), and a Q-Q plot shows a higher occurrence of high concentration outliers. These four events with the highest tritium concentration will be discussed later. The mean tritium value of 4.52 TU measured at Corte over the sampling period compares well to annual averages of GNIP stations in Western Europe and on the shores of the Mediterranean Sea (Figure 2), such as the nearby stations in Girona (4.55 TU), Palma de Mallorca (4.62 TU), Monaco (3.76 TU), and Ninfa (4.69 TU). These observations indicate that measurements of tritium in precipitation over the course of one year produce a distribution of concentration data that statistically represent averaged values for spatial comparison.

Concerning seasonality, one can usually expect a spring to early summer maximum of tritium concentrations in precipitation because of increased moisture exchange from the stratosphere to the troposphere ('tropopause leak'; Martell, 1959; Storebø, 1960). Observed seasonal distribution of tritium concentrations roughly follows this trend (Figure 4). While the highest values were found from April to June, the remainder of the year exhibit depleted tritium concentrations. Outlier events, such as in November do not fit the general concept. However, they rarely occurred and contributed only with small rain amounts. They will be discussed separately in the Sect. 4.3.

[Figure 4 near here]

### 4.2 Comparison of tritium and trajectory-derived parameters

The contribution of moisture from the five defined source regions (Figure 3) confirms some of the initial hypothesis about atmospheric dynamics that probably influence tritium concentrations. The surrounding Mediterranean Sea is the main moisture contributor for the station at Corte. Moisture from Western Europe is contributing less than the Mediterranean Sea. Nonetheless, its contribution is more frequent than from Continental Europe, as a result of the prevailing pressure systems that induce western winds. The Atlantic Ocean plays a subordinate role. This is most probably due to the long travel path of air to the measurement site and rainout on land barriers, such as Western Europe. The continental regions of Africa and Continental Europe do hardly contribute moisture, due to their low moisture capacity when compared to open water bodies. In contrast to tritium concentrations, an obvious seasonal change in moisture sources could not be detected. Such variances were difficult to reveal, because our results rely on calculations of air trajectories of individual events.

In order to advance the knowledge of tritium distribution, tritium event concentrations are compared to different parameters derived from trajectory calculations in the following paragraphs.

The classification of the working area into five tritium source regions was made under consideration of theoretical processes that are thought to shape tritium distribution in the atmosphere as detailed in the introduction. With this, oceanic environments were expected to exhibit very low tritium values in water vapor due to equilibration with ocean water. Over continental parts

of the working area, namely Continental Europe and Africa, the stratospheric source of tritium was expected to lead to an increase in tritium concentrations due to less moisture convection and equilibration with other water compartments, such as oceanic moisture. A comparison of relative moisture contribution of these five source regions with tritium concentration at the measurement site shows low correlations. While linear regressions of moisture contribution against TU of Africa, the Mediterranean Sea, and Western Europe exhibit determination coefficients ($r^2$) of less than 0.06, the Atlantic Ocean shows a negative correlation with an $r^2$ of 0.11 to 0.15, depending on trajectory run time. This is in line with an expectation of lower tritium values in precipitation if substantial parts of the precipitated water originated from evaporated, low tritium ocean water. Moisture contribution from Continental Europe shows a slight positive correlation with tritium concentration ($0.14 \leq r^2 \leq 0.23$). These two regions seem to have the most distinctive influence on precipitation at the measurement site when they contribute moisture to traversing air masses.

The least square regression of water contribution and tritium measurements from Eq. (1) produces estimates for the tritium concentration in these source regions (Table 2). The general trend of expected values is well represented with high values for Continental Europe, where dilution of elevated, stratospheric tritium concentrations with oceanic moisture is inhibited ('continental effect'; Weiss et al., 1979). The calculated value of 7.8 to 8.8 TU is in the range of most GNIP stations in Central Europe. Western Europe shows values of 4.2 to 7.3 TU, whereas the upper value of 7.3 TU was only found for calculations from 10-day-trajectories. This concentration is more than expected from GNIP station values in this region. As Western Europe is positioned between the Atlantic and the Mediterranean, it was expected to be more influenced by oceanic moisture and thus should have lower tritium concentrations, such as shown by trajectory run times of 3 to 7 days (Table 2). This is also reflected by precipitation of the corresponding GNIP stations. One explanation of some higher tritium values in moisture in this region could be anthropogenic influences of tritium-enriched water vapor as a result of nuclear power plant releases. However, these spatially restricted events were not expected to influence water vapor of multiple precipitation events after transport of several hundred kilometers. Further discussion on the influence of nuclear facilities can be found at the end of Sect. 4.3.1. The modelled range of values for Africa of 4.6 to 5.5 TU fits well with observed tritium concentrations in precipitation of the sparsely distributed north African GNIP stations (Figure 2). Due to the 'latitude effect', in which lower latitudes are accompanied by lower tritium concentrations, an even smaller amount of tritium was expected to be typical for precipitation from Africa. One possible scenario for tritium content as high as 5.5 TU could be the low water vapor content in the dry dessert air. As the downward winds above the Sahara Desert contain only minimal amounts of water vapor, even minor contributions of stratospheric tritium could significantly increase the tritium concentration in air moisture. However, as Africa contributes only very small absolute amounts of moisture to the measured precipitation, its relatively high tritium value should not be overrated. Model estimates for contributions from the Atlantic Ocean produce a maximum concentration of 1.4 TU (for 5-day-trajctories) but also negative tritium concentrations. This shows a tendency of the model to assign as small amounts of tritium as possible to the oceanic origin and therefore serves as an indicator of minor tritium contributions. Surface water of the open North Atlantic Ocean is expected to have recent tritium values of around 0.5 to 1.5 TU (Oms, 2018), where the natural background concentration should be between 0.2 TU (pre-bomb concentration; Dreisigacker and Roether, 1978) to 1 TU (Begemann and

Libby, 1957). This is due to the increased vertical exchange of moisture over the ocean, where stratospheric moisture with high tritium content is diluted by oceanic water vapor. Our model results match this expected low concentrations of tritium in the resulting precipitation of water vapor of oceanic origin. The dilution of anthropogenic and natural inputs result in a tritium

concentration of < 2 TU for Mediterranean Sea surface water (Ayache et al., 2015; Roether et al., 2013). Therefore, the modelled values of 4.1 to 4.5 TU for the Mediterranean region are higher than expected from our initial theoretical considerations. This is probably a consequence of this model's calculation procedure where the Mediterranean as the region with the highest moisture contribution (around 60 % on average, Table 2) receives values close to the average of all tritium measurements of 4.5 TU.

In summary, the above shows that assumed tritium concentration tendencies of certain areas are reasonably well represented when calculated from single precipitation events. This is especially the case for areas contributing distinct tritium signals.

To assess the accuracy of the chosen model representation, the modelled tritium values of the source regions can be used to calculate estimates of the measured tritium concentration at Corte (Figure 5). The trend through the estimate-measurement comparison deviates from the 1:1-line with a slope of only 0.21 to 0.27, depending on trajectory run time. This shows that this

simple five-region model can only reproduce around a quarter of the variability of real tritium measurements. One limitation of the calculations of the regionalized tritium model is the assumption that tritium values for selected regions of origin remain constant over time. This is not expected to be the case, due to the seasonality of the tropopause leak that controls the input of tritium from the stratosphere.

[Figure 5 near here]

Compared to the long residence time of tritium in the stratosphere of several years, tritium is removed from the troposphere through rainout in much shorter time (Cauquoin et al., 2016). In order to assess the dynamics of this tropospheric tritium removal, the altitude evolution of trajectories had to be summarized. For each measured precipitation event a moisture contribution weighted median altitude was calculated and compared to the measured tritium concentration (Figure 6). The

325 median is expected to be a more robust representation of a central tendency compared to the mean, because altitude exhibits a non-normal distribution and frequent outliers. The altitude values of trajectory points were weighted by their moisture contribution values ($w_s$) in order to better represent the average altitude of moisture origin. The quality of the fit between these altitude values and their corresponding tritium concentrations (Table 3) was expected to yield information about the dependency of tritium values on air mass altitude history.

[Figure 6 near here]

**Table 3. Coefficients of determination for the correlation of measured tritium concentration against moisture source weighted median altitude (Figure 6) for different trajectory run times.**

| Subset of samples | Trajectory backward run time in days | | | |
|---|---|---|---|---|
| | 3 | 5 | 7 | 10 |
| All samples | 0.03 | 0.10 | 0.07 | 0.10 |
| All samples, excluding tritium outlier concentrations | 0.06 | 0.17 | 0.11 | 0.32 |
| Samples from July to March (no stratosphere leak), excluding tritium outlier concentrations | 0.01 | 0.09 | 0.03 | 0.11 |
| Samples from April to June (stratosphere leak), excluding tritium outlier concentrations | 0.01 | 0.16 | 0.20 | 0.51 |

The above comparison shows a positive correlation of the altitude of moisture origin and tritium concentration for the complete set of measurements, however with a low r² value of 0.03 to 0.10 (Figure 6, solid line). If selected extreme outlier values above 8 TU and below 2 TU are excluded, r² increases to between 0.06 and 0.32 (dot-dashed line). In this study less than 15 % of events are described as outliers and they do therefore not seem to have a strong influence on slope and intercept of the regression (Figure 6) that remain almost the same. Additionally, a seasonal difference in the altitude dependence could be observed. Tritium measurements during the tropopause leak period from April to June exhibited a stronger correlation with trajectory altitude (0.01 ≤ r² ≤ 0.51). During the rest of the year, the slope of the tritium-altitude correlation was smaller and r² was only between 0.01 and 0.11. These observations are in line with measured vertical profiles, where tritium concentrations in tropospheric water vapor generally increased with altitude, especially during spring and summer (Ehhalt, 1971; Bradley and Stout, 1970). Interestingly, determination coefficients improved considerably with trajectory run time (Table 3). This could suggest that longer trajectory run times might better represent tritium contribution to a precipitation event. Initially, a reverse behavior would be expected, because of increased uncertainty and higher variability of integrated altitude signals of longer trajectories. Since the moisture contribution percentages of the tritium source regions and their calculated tritium values do not really change with different trajectory run times (Table 2), tritium concentrations are apparently at least partly uncoupled from moisture uptake.

Overall, the correlation with altitude is better than with any of the other regressors (source regions) discussed above. This general trend seems to imply that high-altitude, high-tritium moisture influences precipitation events, but tracking these processes with back-trajectory models proves to be challenging. However, the events of extreme tritium concentrations that deviate from the trend hint at further influences that have to be taken into account as well.

In short, moisture source altitude dependence of tritium concentrations in precipitation is observed to be strongest during times where vertical input from the stratosphere is increased. Additional influences that are not connected to moisture content and origin seem to play a role also especially during these times.

### 4.3    Tritium outlier events

Six measured precipitation events exhibited notably higher or lower tritium concentrations. All events that have more than 8 TU or less than 2 TU are discussed separately, here. Their corresponding trajectories and its moisture origin are shown on maps and altitude profiles in Figure 7. Additionally, they are marked in Figure 3, Figure 4, and Table 1.

[Figure 7 near here]

#### 4.3.1    High tritium extreme events

With respect to seasonal dynamics, events A to C fall into the period of the tropopause leak. Therefore, elevated tritium concentrations in precipitation are not unexpected. However, the altitude distribution of their trajectories is in contrast to the generally observed trend of increasing tritium concentration with increasing moisture contribution from high tropospheric altitudes. Events A and B both show similar air-mass movements. Event A consists almost exclusively of moisture from

Continental Europe, while event B also must have received moisture from Mediterranean Sea sources. For both events, a majority of moisture uptake happens at lower altitudes of below ~2000 m a.s.l. Although a correlation of altitude with increased tritium concentration could be detected in general and especially in spring and summer (Figure 6), the highest tritium values were measured for these events with low-altitude air mass history. This indicates that either the leakage of stratospheric tritium into the troposphere is too variable to always result in a vertical tritium gradient in the troposphere, or other sources of tritium influence low-altitude moisture trajectories. The significant uptake of low-altitude moisture from Continental Europe points to recycled precipitation and continental waters as a further tritium source. The elevated tritium concentration of precipitation over Continental Europe (Figure 2) thus can be transferred to the next precipitation event elsewhere by re-evaporation and transport.

Event C draws its moisture from trajectories in multiple altitudes, from ground level up to around 3500 m a.s.l. Most of the moisture entered the air mass above the Mediterranean Sea, where low tritium concentrations are expected. A possible explanation for the measured high tritium value of 8.10 TU could be tritium entrainment in high-altitude air (3000+ m a.s.l.), especially in the early parts of some trajectories that collect moisture above Continental Europe. Even minor amounts of moisture from high altitudes could significantly increase tritium concentrations in precipitation (Aggarwal et al., 2016).

Event D shows a tritium concentration of 8.85 TU. In contrast to events A to C, event D did not occur during the tropopause leak in spring to early summer, where tritium concentrations in precipitation are expected to be elevated. This general seasonal trend could be observed in our data, with the exception of event D in November (Figure 4). The trajectories show rapidly traveling air masses that reach the working area boundary within less than four days. Most of the moisture enters the air parcel above the Mediterranean Sea (Figure 3), while secondary moisture uptake happens above the Atlantic Ocean and Western Europe. From these observations and the rather low altitude of trajectory points (<2000 m a.s.l.) only minor amounts of tritium would be expected in the resulting precipitation. This discrepancy can possibly be explained by the addition of tritium by anthropogenic sources. The observed trajectory path leads oceanic air masses over Britain and France, where anthropogenic sources of tritium are present. There is debate which influence such sources of tritium have on their environment. In the near surroundings of nuclear facilities up to several kilometers vapor release was shown to enrich atmospheric tritium concentrations (e.g. Mihok et al., 2016; Chae et al., 2011). The shape and distance of these plumes can vary greatly depending on the type of nuclear facility and atmospheric conditions (CNSC, 2009). With the help of atmospheric dispersion models, plumes were shown to influence tritium concentrations up to 100 km from the release point (Castro et al., 2017). However, farther distances of 200 km and more have yet to be assessed (Connan et al., 2017). In addition to vapor releases, river waters are influenced by liquid discharge from nuclear facilities (e.g. Ciffroy et al., 2006; Jean-Baptiste et al., 2018). Elevated tritium concentrations in river water that discharges into the sea are diluted but can affect tritium concentrations in coastal surface waters. Increased tritium concentrations are found for instance in the English Channel (Masson et al., 2005), the Bay of Biscaya (Oms, 2018), the Celtic Sea (Oms, 2018), and near the mouth of the Rhône (Jean-Baptiste et al., 2018). Exchange of atmospheric moisture with these water bodies with elevated tritium concentrations can increase tritium concentrations in water vapor as well (Connan et al., 2017; Momoshima et al., 1987). Although it seems unlikely that elevated tritium concentrations

in a single precipitation event can be traced back to anthropogenic tritium release over Western Europe, the general concept of anthropogenic influence on the atmosphere is in line with findings of Lewis et al. (1987) who report elevated Tritium concentrations from longer air mass trajectories that originated from Western Europe.

### 4.3.2 Low tritium extreme events

Both low tritium events happen in winter where low tritium concentrations are expected due to the stable separation of stratosphere and atmosphere.

The straight and rapidly moving trajectory of event E hints on low amounts of moisture exchange in Western Europe and over the Mediterranean. The relative moisture contribution inferred from Figure 3 shows a substantial contribution of Atlantic sources that are expected to contain tritium concentrations significantly below 1 TU, and the low altitude of the air parcel above the Atlantic Ocean can explain the matching equilibration with Atlantic moisture. This event seems to be a prime example of parameters that prevent a buildup of tritium in moisture released in a precipitation event.

Event F has calculated trajectories for all 24 hours of the day at which the event took place, because the exact time of precipitation was not logged. Therefore, the exact determination of the moisture origin was not possible. However, since most of the trajectories originate from the Atlantic Ocean, this seems to be a more robust indicator of the low tritium concentration than the supposedly major moisture uptake above the normally relatively dry North Africa. This example underlines the importance of an accurate timeframe for this type of analysis.

## 5    Conclusions

In order to assess moisture sources for precipitation events, air mass back-trajectories were calculated and locations of moisture uptake were derived. The Mediterranean Sea and Western Europe are found to be the prime moisture contributors to measured precipitation events. In conjunction with the measurement of tritium concentrations of these precipitation events and the assumption of regionally similar tritium source values this allowed to identify regions with notable influence on the tritium content. Air masses carrying moisture from Continental Europe and Atlantic Ocean produced distinct high and low tritium concentrations, respectively. A simple model calculated the average of tritium concentrations for each of the source regions. Its results are mostly in line with values expected from generalized tritium distribution processes and measured values at long-term GNIP stations. Although a forward run of the model could only describe around a quarter of the variability of tritium concentrations, this shows that event based data can reasonably reproduce spatial tritium distribution, especially for regions that have distinct source signature. This verifies tritium as a good moisture source tracer for lateral origin. The most probable source of error for the model are seasonal variations in the stratosphere-troposphere exchange that are not included in the calculations. With an increased number of event samples, a simple workaround could be a division of the model into two separate calculations, with and without influence of the tropopause leak.

An obvious seasonal cycle in tritium concentrations could be observed and is in line with expectations of stratosphere-troposphere exchange. Since the input of stratospheric tritium could be detected in precipitation, the question arose if the altitude history of air masses that contributed to an event is correlated with its tritium content. The average moisture source altitude was the best predictor of all tested parameters. Spring and summer events under influence of the tropopause leak exhibited good agreement with moisture source altitude, especially if extreme events were evaluated separately. This strengthens the general concept of altitude dependence of tritium concentrations, not only in ambient water vapor, but also in the resulting precipitation. Additional information from different trajectory runtimes hint on tritium influences that are decoupled from moisture dynamics. Finally, a closer inspection of outlier events revealed possible recycled continental moisture as an important source of tritium to precipitation.

The importance of further research into tritium dynamics in the atmosphere is highlighted by the large variance of tritium concentrations in precipitation when comparing successive events. Considering the supposedly sound knowledge of seasonal tritium distribution, extreme events such as shown in this study can occur all year round. Such events remain mostly hidden when only using the monthly integrated GNIP sampling scheme. Additionally, assessing the contribution of anthropogenic tritium point sources over long distances remains an important issue with respect to tritium concentrations in precipitation events.

## 6    Author contribution

TRJ carried out field sampling with the help of acknowledged colleagues, did all calculations and prepared the manuscript with contributions from all co-authors. JS is responsible for laboratory analyses and considerable manuscript improvement. KT set-up model domain, prepared and converted the ERA5 data for running the HYSPLIT model, and supported atmospheric subjects. FH, EG, and SS helped with field logistics and improved the manuscript. JACB improved the manuscript. RvG helped with field sampling, improved the manuscript, and supervised the project.

## 7    Competing interests

The authors declare that they have no conflict of interest.

## 8    Acknowledgements

This work was supported by the German Research Foundation (DFG) under Grant GE 2338/1-1. The authors would like to thank Martin Häusser, Isabel Knerr, and the whole CorsicArchive team for sampling and field support. We also want to thank the laboratory team of the noble gas laboratory of the IUP Bremen. Additionally, we thank two anonymous reviewers for their helpful comments.

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

**Figures**

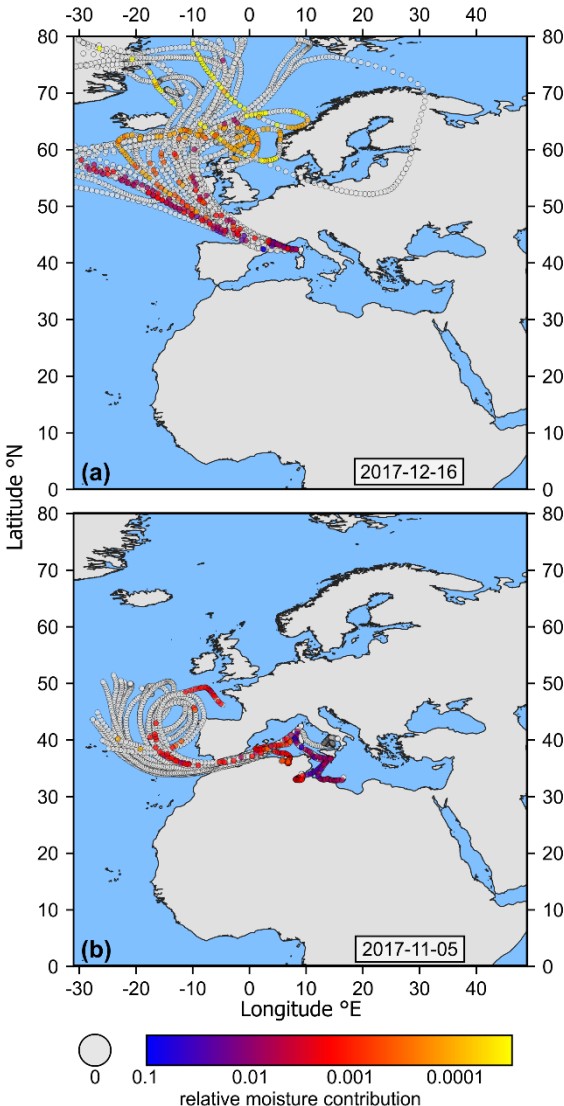

**Figure 1. Example for HYSPLIT backward trajectories of two events on 2017-12-16 (a), and 2017-11-05 (b). The relative amount**
**of moisture uptake of the air mass during movement to the final location is color-coded and sums up to 1. For more details to the air mass movement refer to the text.**

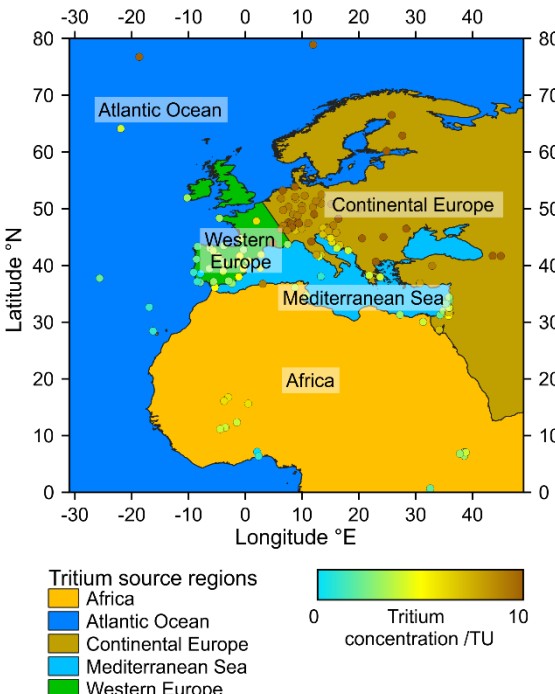

**Figure 2. Map of GNIP stations that provide tritium concentration in precipitation and of the derived tritium source regions.**
**GNIP stations are color coded by the mean of annual averages of tritium concentration over the period of 2000 to 2016 (data coverage of individual stations vary). For more details on differentiation of the five tritium source regions refer to the text.**

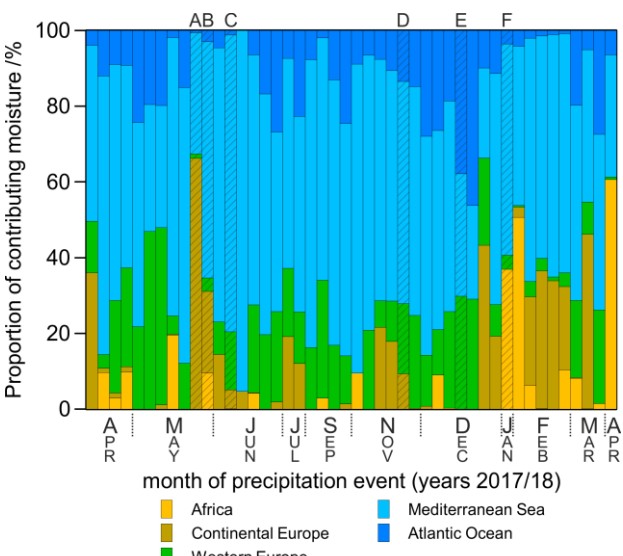

**Figure 3. Relative moisture contribution of each tritium source region to individual precipitation events (depicted as vertical bars) based on 10-day trajectories. Events are listed in chronological order, with their month on the x-axis. Letters on top of the graph correspond to outlier events as detailed in Figure 7 and in the discussion.**


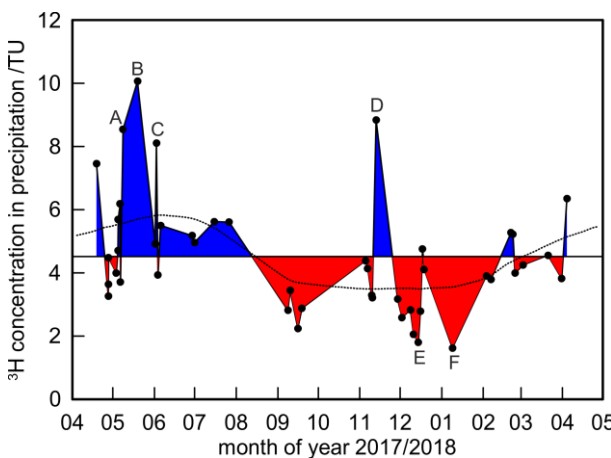

**Figure 4.. Time-series of measured tritium concentration of precipitation events. The horizontal line is the mean of all measured tritium concentrations. Blue areas are above, red areas below the mean concentration. The solid line is a LOESS regression of the measured values.**


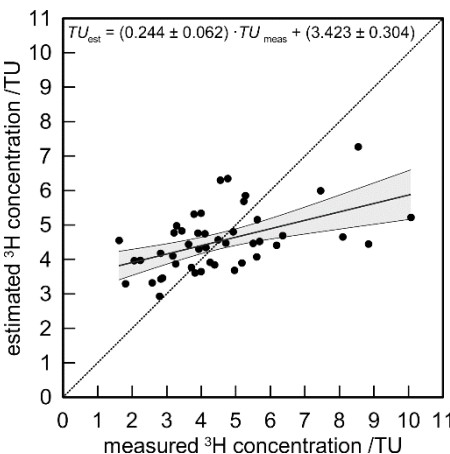

**Figure 5. Comparison between calculated tritium concentrations after Eq. (1) and measured tritium concentrations based on 10-day trajectories. The dotted line is the 1:1 line while the solid line is the linear regression through the data points with the grey band showing the 95 % confidence interval.**


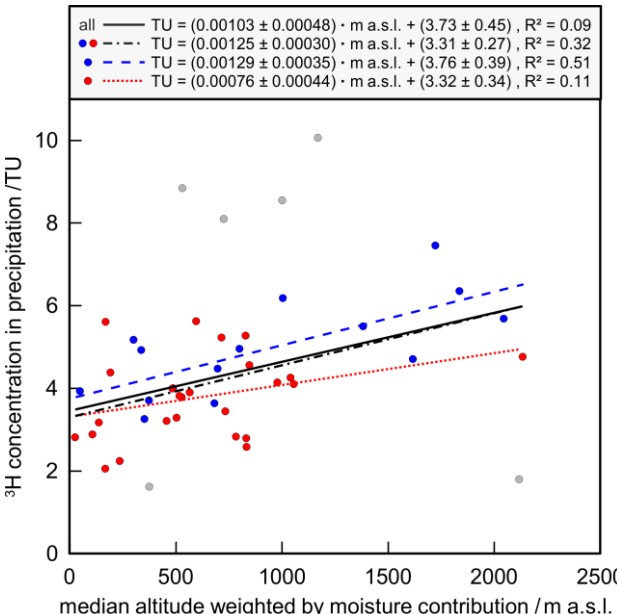

**Figure 6. Tritium concentration of precipitation events plotted against the median of all event-trajectory points weighted by their moisture contribution based on 10-day trajectories. The solid line shows a regression through all data points. The dot-dashed line is the regression through colored data points only, which excludes outliers above 8 TU and below 2 TU. The dashed line is the regression through the blue data points that correspond to events from April to June. The dotted line is the regression through red points from the rest of the year.**

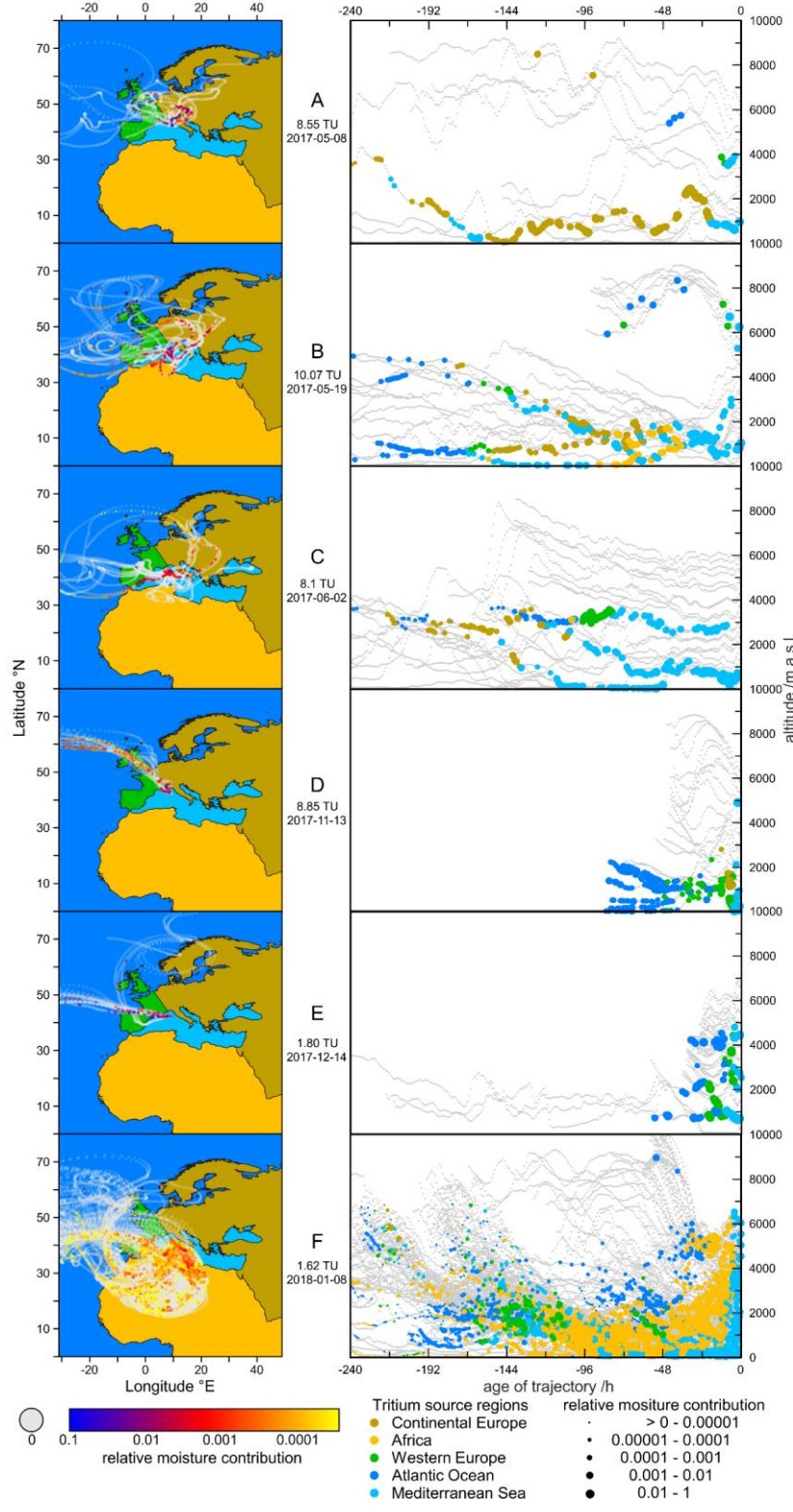

**Figure 7. Maps and vertical cross sections of air mass trajectories for outlier events based on 10-day trajectories. For definition of extreme events refer to Sect. 4.3. Map background colors and cross section trajectory point colors refer to assumed tritium source regions (cf. Figure 2). Colors of map trajectory points and size of cross section trajectory points correspond to relative moisture contribution to the precipitation event (cf. Figure 1).**
