# Peer review of "Tritium as hydrological tracer in Mediterranean precipitation events"

_Atmospheric Chemistry and Physics, 2019_

## Author Comment (AC1) · 27 Sep 2019

The authors like to clarify a section of the methods chapter in order to improve comprehension for readers and reviewers.

We have come to the conclusion that the description of the meteorological ERA5 dataset (Copernicus Climate Change Service (C3S), 2017) and the processing steps we performed that are described in section "2.1 HYSPLIT trajectory model", require some clarification. The actual change in the manuscript will be done in the revision step.

Revised paragraph: "The HYSPLIT model requires input in the form of meteorological grid data. We used the newly available ERA5 grids (Copernicus Climate Change Ser-

vice (C3S), 2017) from the European Centre for Medium-Range Weather Forecasts. This dataset originally has a horizontal resolution of 31 km on a global scale, a vertical resolution of 37 interpolated pressure levels and a temporal resolution of one hour. As input for this HYPLSIT analysis a data subset was extracted with the following specifications: The horizontal and temporal resolutions were left at 31 km and 1 hour. The spatial extent of our model area was clipped to an area of about 40 ° latitudinal and longitudinal distance from our sampling location (0 °N to 80 °N, 31 °W to 49 °E). The upper 6 pressure levels were omitted, leaving 31 pressure levels from 1000 to 20 hPa. Additionally, the model run was cropped in HYSPLIT to an elevation limit of 10000 m a.s.l."

References: Copernicus Climate Change Service (C3S): ERA5: Fifth generation of ECMWF atmospheric reanalyses of the global climate, Copernicus Climate Change Service Climate Data Store (CDS), date of access: 2019-02-06, https://cds.climate.copernicus.eu/cdsapp#!/home, 2017.

---

## Referee Comment (RC1) · Anonymous Referee #1 · 24 Oct 2019

Authors: Tobias R. Juhlke, Jürgen Sültenfuß, Frédéric Huneau, Emilie Garel, Sébastien Santoni, Johannes A. C. Barth, Robert van Geldern

General comments

In their manuscript entitled "Tritium as hydrological tracer in Mediterranean precipitation events" the authors address questions for tritium sources, sinks and transfer processes. They aim at comparing spatial tritium distribution patterns with moisture source regions of discreetly sampled precipitation events. They present isotope data from a one year field sampling campaign, where tritium concentrations of 46 samples of individual precipitation events on the island of Corsica were analyzed and compared to their moisture origin. To better understand short term tritium variations, air mass back-trajectory modelling (HYSPLIT) was applied to identify the sources of water vapor. They could show

that model-derived source region tritium concentrations agreed well with annual mean station values. Overall, the manuscript is well structured and nicely written. The topic fits well to the scope of the journal. I only suggest minor revisions prior to acceptance and publication in Atmospheric Chemistry and Physics.

Specific comments

4.2.2 Low tritium events

Strictly speaking, if the time of precipitation for outlier event F is unknow, modelling and evaluation of the moisture origin is not correct. However, the pattern of the trajectories shows the resulting high uncertainty impressively. Did you check the event on 2017-12-10 (and on 2017-09-15) as well? Their tritium concentrations are also very close to the 2 TU line.

Fig. 3

Outlier events could be better distinguished if pale colors or dotted or dashed symbols for the corresponding events would be used.

Fig. 6

The different regression lines are hard to distinguish. Perhaps scale y-axis from 1 to 11 or enlarge the diagram.

Technical corrections

Blank is missing after the ';' when more than one REF is cited throughout the manuscript, e.g. l. 43, l. 44, l. 70, l. 104, l. 129, l. 240, l. 257, l. 278, l. 317, l. 321, l. 326.

l. 340

'originate' instead of 'originates'.

Please also note the supplement to this comment:
https://www.atmos-chem-phys-discuss.net/acp-2019-725/acp-2019-725-RC1-supplement.pdf
* * *

---

## Referee Comment (RC2) · Anonymous Referee #2 · 18 Nov 2019

The article by Juhlke et al introduces a novel tracer of moisture sources in precipitation that could be possibly used to identify present and past atmospheric circulation patterns to help us better constrain hydrological parameters in climatic models. I find the possibilities introduced by the article promising and worth publishing, however, being the "first of its kind", the methodology and the implications must be better explained in a revised manuscript. I detail below some points of confusion and suggest possible ways of improvement.

Introductory first two paragraphs. This part sounds like a collection of statements on tritium, rather than a coherent introductory text that sets the background of the analysis. The links between different tritium sources and reservoirs are not clearly defined, nor how the constant decay of 3H and lack of supply will impact futures studies. This entire

section should have some more (specific) time dedicated to it.

Study site A brief paragraph on atmospheric dynamics at the study site would be most-welcome. Which are the main large-scale circulation pattern affecting moisture delivery in winter and summer? How does the NAO, AMO and MO affect moisture advection? These should be introduced here, before the discussion.

Methods The methodology seems to be somehow unclear. I understood that rain samples were collected on an event-basis (but it is not clear whether all events were sampled – please clarify) and than a methodology to understand the 3H variability was devised – but the analysis seems to be rather confusing (confused?). HYSPLIT is a very useful tool, but it seems that its application here does not use the entire potential it provides. Severla studies in different parts of the world have shown that moisture resulting in precipitation delivered to a given region is picked-up during the last 2-3 days before the rain event, hence the use of 10-days long trajectories seem useless (especially that these long trajectories were that truncated). Further it, is not clear how the trajectories for 10+ levels were used – perhaps sticking to 1-2 levels (or even one, based on previous data on cloud base ta the site) would have resulted in a lower degree of uncertainty. Next, perhaps detailing the reasoning behind the combination of the three weights would be useful. It is only partly explained and than the reader is referred to the original publication. Being a paper that introduces a novel parameter, I find it useful that the entire methodology is clearly explained and self-sustained.

2.3 Tritium in moisture source regions This part is very confusing. It is not clear how the different regions were delineated> based on 3H values in local precipitation from the IAEA database? Were the values calculated for overlapping periods of time? 3H values change in time and if the analyzed periods were not similar, biases could occur. I suggest reorganize this part (and the subsequent results section) by replacing the examples in fig 1 with trajectories showing moisture pick-up regions independent of tritium measurements (something similar with present Fig. 7, but perhaps for one altitude only; see also Krkelc et al., 2018). I would than use these maps to correlate moisture

pick-up regions with a map of 3H in European precipitation and thus derive theoretical values of 3H, which could be than correlated with the measured values. While this seems to have been attempted, it was done in a very confusing way, bordering circularity in arguments. Separately, a discussion of the measured values in relation to atmospheric circulation during the analyzed period is required. This analysis could result in a potential link between large-scale atmospheric patterns and 3H values and these could be than analyzed against the HYSPLIT-based work to put weight behind "tritium as a hydrologic tracer".

I know these suggestions require a massif reorganization of the paper, but it is my opinion that like this the analyses would make a better use of the data gathered by the authors.

Krklec, K., Domínguez-Villar, D., and Lojen, S.: The impact of moisture sources on the oxygen isotope composition of precipitation at a continental site in central Europe, Journal of Hydrology, 561, 810-821, https://doi.org/10.1016/j.jhydrol.2018.04.045, 2018.

---

## Author Response (AR1)

**Response to Reviewers**

Manuscript ID acp-2019-725

*"Tritium as hydrological tracer in Mediterranean precipitation events"*

Reviewer comments in italic; our answers in normal text.

Changes in the manuscript are highlighted by the MS Word tracking tool

**Referee #1**

**General comments**

*In their manuscript entitled "Tritium as hydrological tracer in Mediterranean precipitation events" the authors address questions for tritium sources, sinks and transfer processes. They aim at comparing spatial tritium distribution patterns with moisture source regions of discreetly sampled precipitation events. They present isotope data from a one year field sampling campaign, where tritium concentrations of 46 samples of individual precipitation events on the island of Corsica were analyzed and compared to their moisture origin. To better understand short term tritium variations, air mass back-trajectory modelling (HYSPLIT) was applied to identify the sources of water vapor. They could show that model-derived source region tritium concentrations agreed well with annual mean station values. Overall, the manuscript is well structured and nicely written. The topic fits well to the scope of the journal. I only suggest minor revisions prior to acceptance and publication in Atmospheric Chemistry and Physics.*

**Specific comments**

1) *4.2.2 Low tritium events*

   *Strictly speaking, if the time of precipitation for outlier event F is unknow, modelling and evaluation of the moisture origin is not correct. However, the pattern of the trajectories shows the resulting high uncertainty impressively. Did you check the event on 2017-12-10 (and on 2017-09-15) as well? Their tritium concentrations are also very close to the 2 TU line.*

We acknowledge that since the exact time of precipitation for event F is not known, the evaluation of moisture sources in this case is less precise than for other events. Aside from the fact that the precipitation time was not logged for this event, long lasting precipitation events will necessarily result in more diverse and fuzzier moisture source determination, because of the spread in trajectory starting times and therefore increased number of trajectories. Nevertheless, the general air movement is roughly constant during this day, with air masses arriving from the Atlantic Ocean, circulating once above Northern Africa and ending up at the field site.

The comparison with further low tritium events (2017-12-10 and 2017-09-15) shows that trajectories for these events also originate at the Atlantic Ocean and have their majority of moisture uptake above oceanic terrain. Since the event time is known for these events, less trajectories were calculated, which leads to a clearer picture.

*Figures:*

*Fig. 3: Outlier events could be better distinguished if pale colors or dotted or dashed symbols for the corresponding events would be used.*

We chose not to change the colors in order to keep them consistent across all figures. Instead we highlighted them by a diagonal hatch pattern.

*Fig. 6: The different regression lines are hard to distinguish. Perhaps scale y-axis from 1 to11 or enlarge the diagram.*

We tried different methods to better visualize the regression lines, including changing axis limits, axis length and plot size. We decided to rescale the y-axis and slightly enlarge the diagram.

**Technical corrections**

1) *Blank is missing after the ";" when more than one REF is cited throughout the manuscript, e.g. l. 43, l. 44, l. 70, l. 104, l. 129, l. 240, l. 257, l. 278, l. 317, l. 321, l. 326.*

We checked the manuscript again and added a blank space when multiple references were cited.

2) *Line 340 „originate" instead of „originates"*

We included the mentioned correction.

**Referee #2**

**General comments**

*The article by Juhlke et al introduces a novel tracer of moisture sources in precipitation that could be possibly used to identify present and past atmospheric circulation patterns to help us better constrain hydrological parameters in climatic models. I find the possibilities introduced by the article promising and worth publishing, however, being the "first of its kind", the methodology and the implications must be better explained in a revised manuscript. I detail below some points of confusion and suggest possible ways of improvement.*

**Specific comments**

1) *Introductory first two paragraphs. This part sounds like a collection of statements on tritium, rather than a coherent introductory text that sets the background of the analysis. The links between different tritium sources and reservoirs are not clearly defined, nor how the constant decay of 3H and lack of supply will impact futures studies. This entire section should have some more (specific) time dedicated to it.*

We rearranged and extended the introduction to have a more comprehensible narrative.

2) *Study site A brief paragraph on atmospheric dynamics at the study site would be most-welcome. Which are the main large-scale circulation pattern affecting moisture delivery in winter and summer? How does the NAO, AMO and MO affect moisture advection? These should be introduced here, before the discussion.*

We included a new paragraph on general atmospheric circulation patterns affecting our field site.

3) *Methods The methodology seems to be somehow unclear. I understood that rain samples were collected on an event-basis (but it is not clear whether all events were sampled – please clarify) and than a methodology to understand the 3H variability was devised – but the analysis seems to be rather confusing (confused?).*

Unfortunately, we were not able to collect samples of each event, but we expect the collected samples to be a representative subset of all events. We included a statement in the methods section that clarifies this issue.

4) *HYSPLIT is a very useful tool, but it seems that its application here does not use the entire potential it provides. Severla studies in different parts of the world have shown that moisture resulting in precipitation delivered to a given region is picked-up during the last 2-3days before the rain event, hence the use of 10-days long trajectories seem useless (especially that these long trajectories were that truncated).*

We agree that the most significant part of moisture for a precipitation event enters the air parcel during the last two to three days. We could also verify this for our data. Nevertheless, multiple important studies also used longer trajectory run times of 10 days (e.g. Visser et al., 2018; Pfahl and Wernli, 2008, 2009). Additionally, the calculation procedure for one of the weights ($w_{incr}$) to determine the location of moisture uptake accounts for the fact that an older moisture source signal can be lost or is mixed with newly added moisture later in the trajectory evolution. We hinted on this in the methods and it is explained in detail in the original publication (Visser et al., 2018). We now updated the explanation in the text in order to make it more comprehensible. Furthermore, we decided to expand our calculations and have now compared moisture source calculations for trajectory run times of 3, 5, 7 and 10 days (e.g. Table 2 and multiple text parts). In short, we found that spatial moisture source and tritium distribution is fairly constant over different trajectory run times. For tritium correlation with altitude, we noticed an improved fit with longer trajectory run times. This may hint on the importance of stratospheric tritium input that is somewhat independent of moisture transport.

5) *Further it, is not clear how the trajectories for 10+ levels were used – perhaps sticking to 1-2 levels (or even one, based on previous data on cloud base ta the site) would have resulted in a lower degree of uncertainty.*

The ideal approach to determine the starting altitude for trajectory runs would be the measurement of the cloud base altitude, as you mentioned. Since we have no actual

measurements of cloud base altitude for our collection site, we chose to use multiple starting altitudes to capture a broader image of atmospheric conditions during precipitation formation. Visser et al. (2018) used six elevation steps from 1000 to 6000 m a.g.l., whereas we increased this resolution to 0 to 6000 m a.g.l. in 500 m steps. Lower altitudes are important because of rapid moisture dynamics in the planetary boundary layer, but high altitudes are also interesting, because of natural tritium input from the stratosphere. The actual moisture contribution from different altitudes is accounted for by one of the calculated weigths ($w_z$). This reduces the noise of having twelve different trajectory starting altitudes, by weighing by the moisture evolution during the last trajectory step. We now updated the explanatory text in the methods section in order to make it more comprehensible. For our calculated trajectories, with 3 as well up to 10 days run time, the starting altitude of 1000 m was the most important moisture contributor.

6) *Next, perhaps detailing the reasoning behind the combination of the three weights would be useful. It is only partly explained and than the reader is referred to the original publication. Being a paper that introduces a novel parameter, I find it useful that the entire methodology is clearly explained and self-sustained.*

The original publication (Visser et al., 2018) needed almost three pages, one figure and eleven equations in order to fully explain their methodology. In order to keep our manuscript concise and focused on our data, we decided to give only a short summary of the calculation for the moisture source weights and then refer to the original publication. We now have updated the explanation of this methodology in our revised manuscript to give more details about the reasoning behind these weights.

7) *2.3 Tritium in moisture source regions This part is very confusing. It is not clear how the different regions were delineated> based on 3H values in local precipitation from the IAEA database? Were the values calculated for overlapping periods of time? 3Hvalues change in time and if the analyzed periods were not similar, biases could occur.*

The tritium source regions were delineated based on actual measured tritium concentrations in precipitation across Europe and Africa, and based on generally assumed tritium dynamics. Essentially, areas above large water bodies (Atlantic Ocean and Mediterranean Sea) were expected not to constitute a source of high tritium concentrations, because of low tritium concentrations in the water body itself. The Mediterranean Sea as a basin with restricted exchange with ocean water was expected to provide slightly tritium-enriched water vapor compared to the Atlantic Ocean. Areas above land were delineated using tritium concentration of monthly precipitation samples in GNIP station of the IAEA. Figure 2 shows a map of these stations with a color code for long-term average tritium concentrations between 2000 and 2016. Just like Cauquoin et al. (2015), we chose 2000 as the starting year for the average in order to exclude high tritium concentrations in precipitation that were influenced by bomb-tritium. 2016 was the last year for which GNIP data was available at the time of our analysis. Western and

Continental Europe were divided by apparent differences in tritium concentration in precipitation (Figure 2).

> 8) *I suggest reorganize this part (and the subsequent results section) by replacing the examples in fig 1 with trajectories showing moisture pick-up regions independent of tritium measurements (something similar with present Fig. 7, but perhaps for one altitude only; see also Krkelc et al., 2018). I would than use these maps to correlate moisture pick-up regions with a map of 3H in European precipitation and thus derive theoretical values of 3H, which could be than correlated with the measured values. While this seems to have been attempted, it was done in a very confusing way, bordering circularity in arguments.*

We thank the reviewer for this suggestion. The mentioned publication also uses trajectories and regionalized maps of origin, albeit for stable isotope analysis of oxygen and hydrogen. Their method is slightly different, where integrated moisture source maps from trajectory data (=footprints) are regionalized and compared with a monthly integrated isotope value. Since we sampled event water, we decided to go one step further and try to assign tritium values to the regions (based on all sampled events, and including a thorough discussion) and then test if these regional values can reproduce single event concentrations. Therefore, we think that this cannot be regarded as circularity in arguments. We also decided against the use of footprints, because on the event scale they contain the same information as the trajectory points themselves. In order to see if these kind of analyses can be carried out on an event-based timescale, we think our calculations are one suitable option. We hope to have eliminated some confusion in this revised version with a clearer methods section and some added details during results and discussion.

> 9) *Separately, a discussion of the measured values in relation to atmospheric circulation during the analyzed period is required. This analysis could result in a potential link between large-scale atmospheric patterns and 3H values and these could be than analyzed against the HYSPLIT-based work to put weight behind "tritium as a hydrologic tracer".*

We preliminary checked for a correlation of the measured tritium concentration in event precipitation with daily and monthly values of the NAO, AO and MO indices, but none could be detected. Therefore, we decided not to focus on these large-scale processes and general atmospheric conditions in this manuscript. But we gladly accept this hint and may consider a follow-up manuscript with an emphasis on large-scale patterns.

> 10) *I know these suggestions require a massif reorganization of the paper, but it is my opinion that like this the analyses would make a better use of the data gathered by the authors.*

We thank the reviewer for his helpful and constructive comments and hope that our corrections and additions to the revised version clarify the applied methods of calculation and improve mentioned weak points of our initial submission.

References:

[revised manuscript text omitted]

---

## Author Response (AR2)

**Response to Reviewer comments 2**

Manuscript ID acp-2019-725
*"Tritium as hydrological tracer in Mediterranean precipitation events"*

Reviewer comments in italic; our answers in normal text.
Changes in the manuscript are highlighted by the MS Word tracking tool

**Referee #2**
**General comments**
*The authors submitted an improved version of their original submission. The issues raised in the original review were addressed and while some of them could not be solved with the existing data, the explanations clearly stated this.*

*Apart from a few technical comments (below) and one on the discussions, I have no other concerns.*

**Specific comments**

1) *Fig. 3 - perhaps it should be made clear in the caption that each vertical bar represents an event*

We clarified this in the figure caption.

2) *Are the averages in fig. 2 calculated for periods of similar length (i.e., do all stations have data covering the full 2000-2016 period)?*

No, every station has different data coverage period. Presented station averages are annual means of monthly means (i.e. in a first step 12 means were calculated, one for each month, in the second step the annual mean of these monthly means was calculated). Cauquoin et al. (2015) argued that a timespan starting with the year 2000 will exclude most of the bomb-tritium influence while including a sufficient amount of samples. We acknowledge that for a finer discussion of local and seasonal differences in Tritium content one would have to apply a more sophisticated data selection process. We checked if exclusion of certain years or stations with low data count changes the general trend over our working area, but could not find substantial differences. Therefore, we think our use of the data is valid, since we only used these GNIP averages as a coarse proxy to delineate the tritium source regions.

3) *Fig. 6. perhaps the regression lines could be color coded, as well?*

We colored some regression lines to be more easily identifiable.

4) *Sampling: the authors sampled 42 events and state that "This number is a subset of all precipitation events that is expected to be representative of the total rainfall during the study period". It is not clear however how much of the rain falling during the study*

*period has been sampled? How as representativeness determined? This should be clarified (in terms of volume percentages, perhaps?).*

Around 43% of the total precipitation amount during the one-year period (mid-April 2017 to mid-April 2018) was sampled. We think the sampled subset adequately represents the "total" precipitation, concerning rain intensity and seasonal distribution (see Figure R1).

[Figure]

Figure R1: Proportions of sampled rain hours

5) *Conclusions vs discussions vs. response to reviewers.*
   *by reading the response and the manuscript, it looks that the later is somehow less straightforward, with the general direction of the discussions being lost in the amount of data (measured and modeled). Further, the conclusions don't make justice to the discussions: they should be streamlined to better reflect the findings. For instance this paragraph form the response very nicely summarizes the findings, yet it is not expressed as such in manuscript. I suggest structure chapter 4.1 around 2-3 research questions/hypothesis and than resolve them and end the respective paragraph with a statement on the findings that should be further used in the "conclusion" (e.g., role of stratospheric input should be discussed in one single paragraph, not spread through the discussion of the regional sources).*

We substantially rearranged the discussion section and used more subheadings to give the reader a more structured narrative. In addition, we added short summaries at the end of paragraphs in the discussion and made other improvements to better inform about outcomes pertaining to the research questions. Additionally, we rewrote a large part of the conclusions to be more focused and give summarizing answers on the research questions.

References:

[revised manuscript text omitted]